# Curated Database and Preliminary AutoML QSAR Model for 5-HT1A Receptor

**DOI:** 10.3390/pharmaceutics13101711

**Published:** 2021-10-16

**Authors:** Natalia Czub, Adam Pacławski, Jakub Szlęk, Aleksander Mendyk

**Affiliations:** Department of Pharmaceutical Technology and Biopharmaceutics, Jagiellonian University Medical College, 30-688 Kraków, Poland; natalia.czub@doctoral.uj.edu.pl (N.C.); j.szlek@uj.edu.pl (J.S.); aleksander.mendyk@uj.edu.pl (A.M.)

**Keywords:** 5-HT1A receptor, curated database, QSAR, Mordred descriptors, AutoML, pKi, SHAP

## Abstract

Introduction of a new drug to the market is a challenging and resource-consuming process. Predictive models developed with the use of artificial intelligence could be the solution to the growing need for an efficient tool which brings practical and knowledge benefits, but requires a large amount of high-quality data. The aim of our project was to develop quantitative structure–activity relationship (QSAR) model predicting serotonergic activity toward the 5-HT1A receptor on the basis of a created database. The dataset was obtained using ZINC and ChEMBL databases. It contained 9440 unique compounds, yielding the largest available database of 5-HT1A ligands with specified pKi value to date. Furthermore, the predictive model was developed using automated machine learning (AutoML) methods. According to the 10-fold cross-validation (10-CV) testing procedure, the root-mean-squared error (RMSE) was 0.5437, and the coefficient of determination (*R*^2^) was 0.74. Moreover, the Shapley Additive Explanations method (SHAP) was applied to assess a more in-depth understanding of the influence of variables on the model’s predictions. According to to the problem definition, the developed model can efficiently predict the affinity value for new molecules toward the 5-HT1A receptor on the basis of their structure encoded in the form of molecular descriptors. Usage of this model in screening processes can significantly improve the process of discovery of new drugs in the field of mental diseases and anticancer therapy.

## 1. Introduction

Serotonin (5-HT, 5-hydroxytryptamine) is mainly produced and present in the peripheral tissues. Only 5% of this monoamine occurs in the brain [1]. Serotonin interaction with various receptors leads to modulation of numerous physiological and pathological processes, in both the peripheral and the central nervous system (CNS). Dysregulation of the serotonergic system causes many psychiatric and neurological disorders, such as migraine, pain, anxiety, schizophrenia, movement disorders, and depression [2,3]. Among these, the latter affects over 300 million people worldwide, irrespective of gender, race, or socioeconomic differences. Untreated depression may lead to deterioration of health and quality of life and, in the advanced form of disease, to suicide [4]. There are currently 14 distinguished subtypes of serotonin receptors, of which the 5-HT1A receptor is the most researched and described [5]. This receptor plays an important role in neuronal activity functions (muting neural transmission, synaptic plasticity, neurogenesis, and neuroprotection) and in changes of behavior, as confirmed by animal studies (reduction of anxiety, depression, and pain) [1,3]. Moreover, the 5-HT1A receptor may become a target for the anticancer treatment of colorectal cancer, small-cell lung cancer, bladder cancer, and prostate cancer. The antitumor effect may be mediated by antagonism of 5-HT1A receptors through blocking serotonin-induced tumor growth [6].

Discovery of compounds with specific activity is a very time-consuming and costly process [7]. The efficacy of introduction of a drug to the market is very low. It is estimated that the success rate of a clinical trial is a few percent. According to the data gathered by Wong et al., the overall success rate, depending on the disease, sponsor, biomarker presence, lead indication status, and time, can vary from 1.2% to 8.3% [8]. Therefore, it is paramount to employ every available technique to increase drug development efficiency. A field of science that effectively accelerates the drug discovery and development process is artificial intelligence (AI) with a specific branch that includes machine learning (ML). By employing ML tools to available data, both the activity of compounds and their physicochemical properties can be predicted. A specific class of model describing the chemical structure of binding compounds with their biological activity is called a quantitative structure–activity relationship (QSAR) [7].

In the current state of knowledge, contemporary QSAR models for 5-HT1A receptors mainly represent classification tasks. Thus, the threshold-based distinction between inactive and active compounds or between agonists and antagonists of the 5-HT1A receptor is at scope without any concern of the strength of this activity [2,9,10]. As this approach is still valuable for the drug discovery phase, more sophisticated regression-based models are certainly desirable for their ability to predict potency of a future drug candidate. A few studies of regression models were based on a specific group of compounds in terms of their chemical structure, e.g., arylpiperazinthioalkyl or thienopyrimidinone derivatives. The target value was Ki—a continuous variable representing a compound’s affinity to the receptor. The tested compounds also influenced other types of receptors, α1-adrenergic and 5-HT1B, respectively [11,12]. Among the studies in which the compounds act only on the 5-HT1A receptor and the affinity value was predicted, only arylpiperazine derivatives appeared [13,14,15]. To our best knowledge, there are no Ki/pKi value prediction studies with use of a database containing 5-HT1A ligands diverse in terms of both structure and physicochemical properties.

The aim of our work was to obtain the most suitable model to predict compound activity for the 5-HT1A receptor, based on the largest possible curated database acquired during this research. For the output, we chose pKi (negative logarithm of inhibition constant Ki) as a measure of ligand affinity to the target [16]. For modeling, we used the automated machine learning tool (AutoML) from the H_2_O AutoML platform [17]. Our secondary objective was to explore the possibility of fully automated model development in the area of drug discovery, as AutoML was employed for both feature selection and final model development. In the area of cheminformatics, compounds can be represented by various types of graphs, descriptors, or image-based representations as input data to create a predictive model [18]. In our research, we chose QSAR models based on standard molecular descriptors.

In the following sections, we describe the processes of database development, curation, and analysis. Furthermore, QSAR modeling with AutoML is depicted including feature selection and search for a final QSAR model, together with its analysis via an explainable AI approach.

## 2. Materials and Methods

### 2.1. Database

In this study, we used two publicly available databases, ChEMBL and ZINC. Among compound characteristics, the databases contain data on the bioactivity, which are regularly extracted from scientific articles [19,20].

The databases were downloaded and processed offline with the use of various tools such as DataWarrior, Python library Pandas, and RDKit [21,22,23].

Data processing introduced a merging procedure of two databases, checking for errors in data (i.e., non-numerical values), and checking for missing values and duplicates of molecules between two source databases, i.e., ChEMBL and ZINC.

The next step in building the database was to add molecular descriptors to the corresponding pKi values. Molecular descriptors are defined as a result of mathematical procedures that transform the chemical information encoded in the symbolic representation of a compound into a useful numerical value. The Mordred package was used to calculate descriptors due to its high computational efficiency and stability [24]. In our research, we used two-dimensional (2D) descriptors to avoid the impact of compound optimization and parameters related to the three-dimensional properties of molecules. Mordred was executed with the Python 3 environment under a Linux operating system.

### 2.2. AutoML Model

This computational experiment was designed to develop a multiple-input/single-output (MISO) model employing automated machine learning tools (AutoML). The inputs were abovementioned molecular descriptors calculated with the use of the Mordred package, and the output of the model was the pKi affinity value for the 5-HT1A serotonin receptor. AutoML tools can develop complex empirical models with high efficiency and at reduced computational time, while maintaining good quality. In this study, we used H_2_O AutoML with a Python interface [17]. In order to further automate the process, we wrote a Python script [25] devoted to combining four basic stages of modeling: feature selection, algorithm selection, model development, and parameter optimization. A scheme of the machine learning workflow applied in this experiment together with AutoML settings is presented in Figure 1. All stages of the presented workflow are done automatically. The user provides only the preprocessed dataset and settings for AutoML, such as computation time, number of *n*_try, and feature selection threshold.

At every stage, we used all available algorithms in the H_2_O implementation of AutoML, i.e., distributed random forest (DRF), extremely randomized trees (XRT), generalized linear model (GLM), extreme gradient boosting machine (XGBoost), gradient boosting machine (GBM), deep learning (fully connected multilayer artificial neural network), and stacked ensemble models. Ensemble models are created in two ways: one ensemble contains all the models, and the second ensemble contains just the best-performing model from each algorithm class/family. Therefore, the “best of family” stacked ensemble model was optimized for production use since it only contains six base models. After specifying a list of base models, a metalearner was trained and tested. During computational experiments, the standard GLM algorithm was used with non-negative weights and Lasso or Elastic regularization as a metalearner. Optimization of the model parameters was done via a random search of hyperparameters [26].

Production model performance was evaluated according to 10-fold cross-validation (10-CV) and expressed by two goodness-of-fit metrics: root-mean-square error (RMSE) and coefficient of determination (*R*^2^). For reference, please see Equations (1) and (2).
(1)$$RMSE=\sqrt{\frac{\sum_{i=1}^{n}{\left(pred_{i}-obs_{i}\right)}^{2}}{n}},$$
where *obs_i_*, *pred_i_* are the observed and predicted values, *i* is the data record number, and *n* is the total number of records.
(2)$$R^{2}=1-\frac{SS_{res}}{SS_{tot}}=1-\frac{\sum_{i=1}^{n}{\left(pred_{i}-obs\right)}^{2}}{\sum_{i=1}^{n}{\left(obs_{i}-obs\right)}^{2}},$$
where *R*^2^ is the coefficient of determination, *SS_res_* is the sum of squares of the residual errors, *SS_tot_* is the total sum of the errors, *obs_i_*, *pred_i_* are the observed and predicted values, and *obs* is the arithmetical mean of observed values.

The Shapley Additive Explanations method (SHAP) [27] was applied to the best developed model to assess the influence of variables on the pKi value. The general procedure behind SHAP calculation is related to the theory of cooperative games developed by Lloyd Shapley in 1953. Shapley values are used for explanation of the predictive model, especially in machine learning [28]. According to the theory introduced by Shapley, if a group of individuals cooperate and form a coalition to achieve a particular goal, e.g., winning the game or finalizing a project, the payoff for every participant should be proportional to their marginal contribution. Proper calculation of each individual’s contribution to the final results should be efficient, symmetrical, and additive. A capability to detect individuals with zero contribution is also required. The solution which meets all the above criteria is called the Shapley value. The marginal contribution for each individual is calculated by generating all permutations of individuals and their results obtained by the formed coalition. Implementation of this theory in the machine learning area leads to the variables treated as individuals creating a coalition, with the predictions of the model as the outcome of cooperation. The question that SHAP analysis answers is how much and in which direction each variable influences the model outcome. The results from the analysis provide a ranking of variables based on the absolute average SHAP value, and they allow finding functional relationships between variables and model predictions.

Computations were executed on a grid system composed of 29 workstations (542 threads) working under the Linux openSUSE Tumbleweed operating system.

## 3. Results

### 3.1. Bulding Database

Using the available ZINC and ChEMBL databases, we obtained a curated database containing 9440 unique ligands of the 5-HT1A serotonin receptor.

In the first step, the two databases were downloaded separately (November 2020). After removing duplicates, the ZINC database contained 8025 and the ChEMBL database contained 3624 unique molecules, respectively. We created a third database derived from the two previously mentioned databases. The total number of compounds was 11,649. However, there were identical compounds found in both ZINC and ChEMBL databases. The total number of duplicates in the combined databases was 3994. The way of handling duplicates depended on pKi differences between the respective compounds from the ZINC and ChEMBL databases ranging from 0 to 4.46. Duplicated compounds with a pKi difference greater than 0.1 were removed completely from the pooled database (212 pairs of molecules). For the duplicates with a pKi difference less than or equal to 0.1, the compounds from the ZINC database were selected. Ultimately, we obtained a database with 9440 unique compounds. The development of the curated database of 5-HT1A ligands is shown in Figure 2.

### 3.2. Exploratory Analysis of the Curated Database

#### 3.2.1. Statistical Exploratory Analysis

The 5-HT1A receptor activity is reported as a negative logarithm of the inhibition constant in the range from 4.2 to 11.0 [16]. The pKi distribution is symmetrical and resembles a normal distribution, which is the most common distribution found in nature. The distribution of pKi values is shown in the histogram below (Figure 3). The Gaussian distribution supports the statement that the represented database, containing many compounds, was suitable for modeling the 5-HT1A-mediated serotonergic activity of various compounds.

According to the calculated Tanimoto similarity coefficient by the RDKit package, the similarity of the compounds to each other was within the range of 0.1553–1.0 for the curated database (median = 0.37). The pairs of compounds with a similarity equal to 1 are considered stereoisomers [23]. Figure 4 illustrates the distribution of the similarity value from the Tanimoto coefficient for 5-HT1A receptor ligands.

The distribution of Tanimoto similarities supports the claim that the molecules present in the database were well differentiated.

#### 3.2.2. Lipinski’s Rule of Five

The classical work from 1997 by Lipinski et al. introduced four basic principles, known as Lipinski’s rule of five, used to ensure the drug likeness of a compound. Following these principles, it is likely to achieve good absorption and permeation through biological membranes. Compounds that meet these principles are characterized by a molecular weight below 500 Da, logP value below 5, and a maximum number of five hydrogen donors and 10 hydrogen acceptors [29].

Based on these features, we investigated how our curated database of 5-HT1A ligands met these expectations. The graphs below show the distribution of compounds in terms of the abovementioned features and pKi value (Figure 5). The orange color shows for which compounds the rule of Lipinski was fulfilled, whereas the blue color shows the opposite situation.

In the work of Veber et al. two more rules appeared, defining a value for polar surface area (PSA) equal to or less than 140 Å^2^ and a number of rotatable bonds below 10 [30]. As for Lipinski’s rule, Figure 6 shows the compounds in our curated database that met the above thresholds in orange, and the compounds that exceeded the designated PSA and rotatable bond limits in blue

In Table 1, we show the percentage of compounds in the curated database which fulfilled each individual rule of five, and how many compounds met all rules. Almost 82% of compounds in the curated database met Lipinski’s rules, and nearly 79% of compounds satisfied Veber et al.’s rules.

The distributions of pKi values for compounds that satisfied the rule of five and for the compounds that did not also had symmetric distributions, suggesting a normal distribution (Figure 7).

The correlation matrix for the six chemical descriptors (MW, nHBDon, SLogP, nHBAcc, TopoPSA, and nRot) and pKi with Pearson’s correlation coefficients are depicted in Figure 8. From the selected features, only the polar surface area (TopoPSA) and number of H-bond acceptors (nHBAcc) had a strong positive correlation.

#### 3.2.3. Drugs Affecting 5-HT1A Receptor

Currently, many ligands of 5-HT1A serotonin receptors are used in the treatment of diseases such as schizophrenia, depression, anxiety, and insomnia. Drugs used in hypertension, arrhythmia, and Parkinson’s disease have multi-receptor profiles also showing affinity for the 5-HT1A receptor. Examples of such drugs are presented in Table 2, along with values of affinity for the 5-HT1A receptor and whether they met Lipinski’s rules. The only drug that did not meet Lipinski’s rules was bromocriptine, whose molecular weight exceeds 500 Da. Other medicines satisfied Lipinski’s principles (LogP, number of H-donors and H-acceptors, PSA, and number of rotatable bonds). The range of pKi values for the presented drugs was within 5.57–9.70 (median = 7.54). The listed examples of drugs can be found in our curated database of 5-HT1A [31,32,33,34,35,36,37,38,39,40,41,42,43,44].

#### 3.2.4. Molecular Descriptors

The first set of obtained descriptors using the Mordred package contained 1613 variables. Missing data were replaced with mean values of the respective column representing the molecular descriptor (a feature). Categorical true/false variables were replaced with 1/0, respectively. Empty columns and those with a maximum value equal to the minimum value (constant columns) were removed. Finally, 1287 input variables of 2D descriptors were obtained.

Mordred 2D descriptors provide information on compounds, such as basic information of molecules (molecular weight, number of individual types of atoms, types of bonds, degree of hybridization, spectral diameter, detour index, number of hydrogen donors and acceptors, molecular distance edge between different types of atoms, polarizability of atoms and bonds, and topological polar surface) and other features derived from a symbolic representation (Zagreb index, adjacency matrix descriptors, Moreau–Mroto descriptors, Moran coefficients, Geary coefficients, and descriptors describing the Burden matrix and Barysz matrix) [45]. Figure 9 shows the distribution of values for some Mordred 2D descriptors for the curated database of 5-HT1A receptors.

No significant linear correlation between pKi and molecular descriptors was found. Pearson correlation (*r*) ranged from −0.197 to 0.200. This indicated a need for nonlinear modeling tools, i.e., a machine learning approach.

### 3.3. Model

The computational experiment allowed us to obtain a model with RMSE equal to 0.5437 and a coefficient of determination (*R*^2^) of 0.7443. The type of model was a stacked ensemble containing 342 models: two XRTs, 49 XGBoosts, two GLMs, 150 GBMs, two DRFs, and 137 deep learning models.

H_2_O AutoML selected the most important descriptors to create a predictive model. From the initial 1287 descriptors, the feature selection stage reduced the number of features to 216. The three most important descriptors for the model were AATSC5d, ATSC4d, and SpMAD. The twenty most important variables are listed in Table 3. A complete list of features selected from the original database is provided in Appendix A.

As a point of reference, a linear model was established using the R environment [46]. According to the 10-CV method, the linear model RMSE was 0.89 and *R*^2^ was 0.32. The model was created on the basis of the input vector consisting of 216 input variables previously selected by the AutoML system. No other linear data analysis method could provide a better model with a comparable number of input variables.

Graphs of the relationship between the observed and predicted pKi values for the 5-HT1A receptor using the two methods mentioned above are shown in Figure 10. Due to the complexity of the examined problem and underlying nonlinearities, the linear model was characterized by worse goodness-of-fit parameters and led to predictions less precise than the AutoML model.

To understand how individual variables affect the pKi values, the Shapley Additive Explanations (SHAP) method was applied [47,48]. SHAP values were estimated on the basis of a subset of 10% randomly chosen records from the database. Figure 11 presents results of the SHAP value calculated for the 10 variables with the highest impact on model predictions with order according to descending absolute average SHAP value (range: 0.07 for SdO to 0.05 for AATS6dv).

A brief analysis of Figure 11 shows that higher values of variables such as SdO, JGI4, PEOE VSA9, ATSC4i, ATSC6i, and AATS6dv negatively influenced the pKi value predicted by the model (SHAP value < 0), whereas variables such as SaaaC, IC2, PEOE VSA10, and SsssN positively influenced the predicted pKi value (SHAP value > 0).

A More detailed analysis of SHAP and variable values revealed interesting relationships. According to the presented results, the most important feature is SdO which represents the sum of dO (=O) E-state of the molecule. The value of the SdO descriptor depends on the presence and location of carbonyl groups in the molecule. If the molecule contains more carbonyl groups, there are more possibilities for water to form hydrogen bonds, and the overall hydrophilicity is increased. It is noticeable that the increased value of the SdO descriptor resulted in decreased pKi values (Figure 12). This trend was observed through the whole domain of this variable. However, regarding the molecules which had a value of SdO = 0 (lack of carbonyl groups in a molecule), the positive influence on pKi was weak (around 0.1). This may suggest that, in the case of 5-HT1A receptors, the surrounding region of the binding site is more lipophilic than the binding site itself. In consequence, more lipophilic molecules have easier access to the binding site. These findings are in line with work of Bondensgaard et al., who suggested the presence of hydrophobic pockets located near the hydrophilic binding site of the 5-HT1A receptor [49]. The hydrophobic pocket consists of aromatic amino acids Trp 358, Phe 361, and Tyr 390, and is in proximity to Asp 116, responsible for ligand–protein binding [50].

In the case of JGI4, representing the topological charge of the molecule and measuring the charge transfers between atom pairs [51], it was observed that values below 0.035 were associated with an increased value of pKi predicted by the model. On the other hand, JGI4 values above 0.035 decreased the predicted value of pKi (Figure 13). Moreover, there was a visible trend suggesting a functional relationship of pKi and JGI4. According to Nowaczyk and Kulig, a decrease in the JGI4 value is observed in molecules which have unevenly distributed electrostatic charge. In other words, JGI4 value is lower for unsubstituted compounds than for compounds substituted with groups of high electrostatic charge (–OH, –O–CH_3_, and –Cl). Therefore, in order to increase pKi value, it suggests the need for a unique charge distribution of the chemical compound [52]. This finding additionally supports the structure of the 5-HT1A receptor suggested by Zlatović et al., in which an acidic amino acid (Asp 116) is located at the ligand–protein binding site [51].

## 4. Discussion

The current scientific literature describing datasets of ligands for 5-HT1A receptor is mainly focused on data from the ChEMBL database. The data from this database were integrated into our curated database.

The research of Warszycki showed a set of 5-HT1A ligands, in a total number of 5039 compounds, divided into training (3616 molecules) and testing (1423 molecules) datasets. Both datasets were retrieved from the ChEMBL database. The aim of this study was to explore a new approach to pharmacophore screening involving the use of an optimized linear combination of models. The authors used three methods of clustering compounds available in the ChEMBL database: using a 3D pharmacophore based on a fingerprint (P3D), MOLPRINT 2D based on a fingerprint (M2D), and the classic manual method (grouping compounds according to a common core). The experiments proved that an automatic method of hierarchical clustering (based on the MOLPRINT 2D fingerprint) is a good option for screening [53]. The abovementioned compounds were also included in the curated database of 5-HT1A receptor ligands.

The second-largest database was presented in the work of Ma and contained 1697 molecules. A novel machine learning-based ligand classification algorithm was introduced: Ligand Classifier of Adaptively Boosting Ensemble Decision Stumps (LiCABEDS). The authors conducted computational experiments using data of four types of serotonin receptors (5-HT1A, 5-HT1B, 5-HT1D, and 5-HT4). The dataset for the 5-HT1A receptor contained 1102 agonists and 595 antagonists. The data were taken from the GLIDA database [54]. The authors created an efficient classifier for antagonists or agonists for different types of serotonin receptors. The database used in this project is not available; thus, it was not possible to compare whether the compounds contained in the GLIDA database were also present in the curated database.

Another database containing data for 5-HT1A came from the work of Kurczab. The aim of this study was to create a classifier identifying target selectivity for 5-HT7 and 5-HT1A receptors. The number of compounds with a specified pKi value affecting the 5-HT1A receptor was 722, retrieved from ChEMBL database. The support vector machine method was used to create the classifier [55]. The created algorithm predicted the selectivity for 5-HT7 and 5-HT1A receptor ligands with high accuracy. The ligands of 5-HT1A receptors used to create the database were included in the curated database.

The fourth set was a database of 346 ligands which were the author’s own experimental data and data collected from the literature. Kuz’min divided the obtained set into four categories in terms of pKi value. These categories became the basis of a classification algorithm that successfully matched new compounds to one of four 5-HT1A receptor affinity groups. The classification and regression trees (CART) algorithm was used for modeling [56]. The obtained model was able to determine the level of affinity of the new compound to the 5-HT1A receptor. However, due to the small number of ligands, the obtained algorithm was adapted to the data; therefore, it would be ineffective for compounds with a different chemical structure.

Weber et al. introduced a database containing information about 88 arylpiperazine derivatives and their affinity to the 5-HT1A receptor. The authors presented research where the prediction of affinity was based on 3D pharmacophore and calculations based on comparative molecular field analysis (CoMFA). The training set consisted of 70 molecules, and the model was externally validated with a test set of 18 compounds [14]. To compare the obtained results with our work, we calculated the RMSE and *R*^2^ for the pKi value predicted by CoMFA. In this case, RMSE was equal to 0.4 and *R*^2^ = 0.8232. The same dataset was used by Veselinović et al. They used the simplified molecular input-line entry system (SMILES) as a molecular representation of arylpiperazine derivatives. QSAR model was developed using CORAL software. Four train–calibration–test splits were performed. Test sets again contained 18 compounds; however, data records were carefully chosen to ensure the same coverage of pKi values in all datasets. The authors used the coefficient of determination (*R*^2^) as a good predictability measure, which, for each split, was above 0.925, and the RMSE was in the range from 0.225 to 0.311 [13]. Jia et al. used the same test set as Weber et al. with 18 compounds. As a molecular representation, they used norm index descriptors (derived from, e.g., a distance matrix and Euclidean spatial distance matrix) and accordingly developed a model for predicting 5-HT1A receptor affinity. The results for the test set were an RMSE equal to 0.3531 and *R*^2^ above 0.866 [15].

Among the abovementioned studies, continuous output (pKi) was used only for the database containing arylpiperazine derivatives; therefore the authors developed regression models that predicted the pKi value [13,14,15]. The limited number of compounds of a similar chemical nature allowed the authors to tune models precisely to the data, which highly improved the model generalization.

Our curated database contains 100× more molecules of different chemical types and properties. The research material of our study, in comparison with the datasets presented above, is the largest available database gathering information on 5-HT1A receptor ligands and counts 9440 unique compounds. In our structure-agnostic approach, unknown structures of various chemical nature may be tested, yet the overall predictability of the model is sacrificed. On the contrary, specialized models with much better predictive ability are characterized with a narrower scope.

## 5. Conclusions

The ongoing development of computational resources and the increase in ML/AI tools enable a significant improvement of the drug discovery and development process. In order to fully exploit this opportunity, large amounts of good-quality data organized in proper form are required. The presented study is focused on the serotonin receptor 5-HT1A, which is a therapeutic target in many CNS diseases (depression, schizophrenia, anxiety, and cognitive disorders), and which might be utilized in the treatment of serotonin-dependent neoplastic diseases in the future [1,3,6].

As a result of the research, a curated database was developed through merging and careful analysis of data available in two leading data sources: ZINC and ChEMBL. Currently, it is the largest and the most diversified freely available database describing molecules and their activity against the 5-HT1A receptor. Moreover, the database is constructed in a way that enables its direct use in predictive model development using ML/AI tools. Such a possibility was also presented, and the model developed with AutoML tool showed good predictive ability with RMSE = 0.5437 and *R*^2^ = 0.7443. To our best knowledge, it is the first predictive model for pKi values for the 5-HT1A serotonin receptor based on such a large and diverse database. The use of this model in screening processes might significantly improve the process of searching for new drugs in the field of mental diseases and anticancer therapy.

Future work will include both database extension and improvement of the QSAR model in terms of predictability and interpretability. For the latter, large-scale calculations will be commenced to find a model with predictability on the level of the model presented in this work (if not improved) with fewer than 216 inputs.

## Figures and Tables

**Figure 1 pharmaceutics-13-01711-f001:**
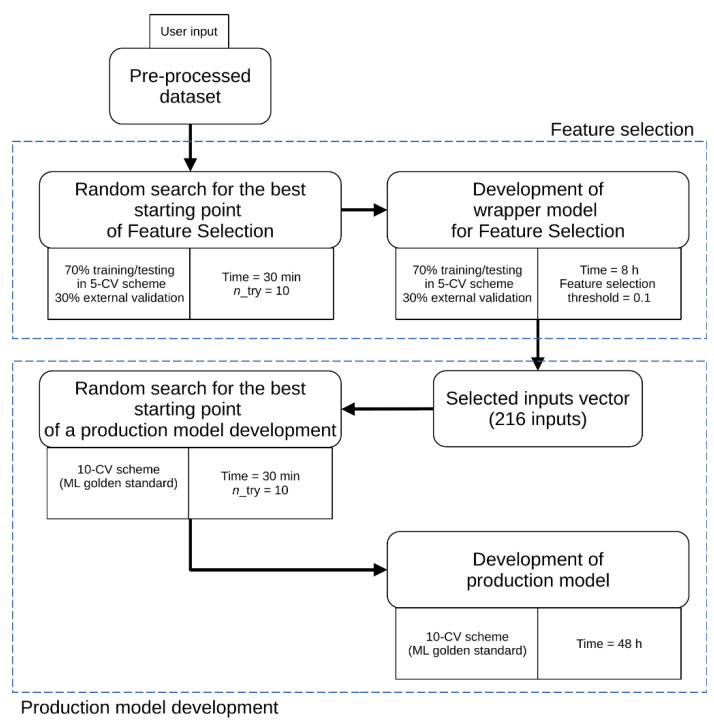
Scheme of applied AutoML H_2_O workflow.

**Figure 2 pharmaceutics-13-01711-f002:**
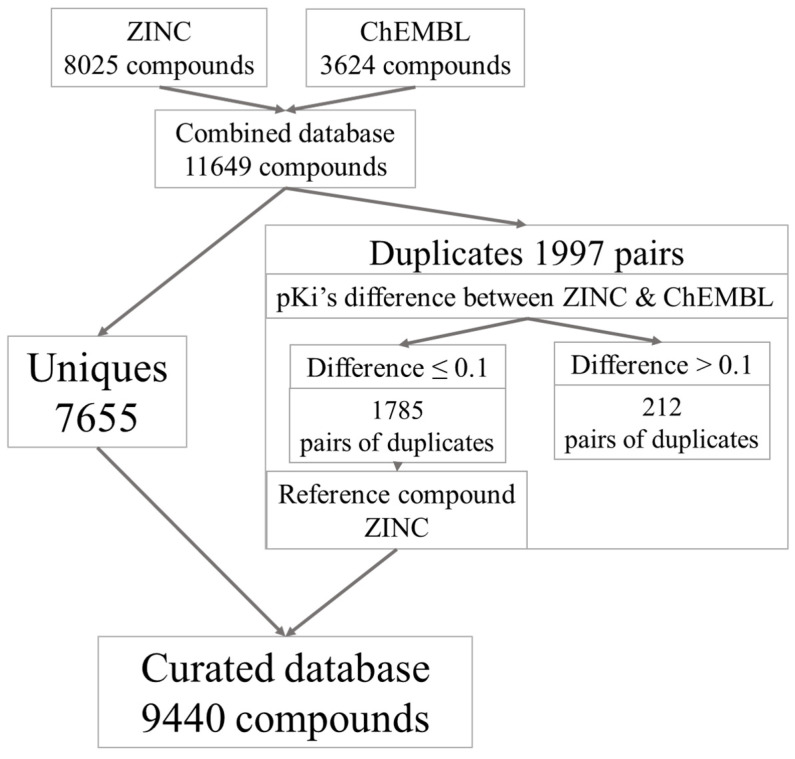
Creation of the curated database of 5-HT1A receptor ligands.

**Figure 3 pharmaceutics-13-01711-f003:**
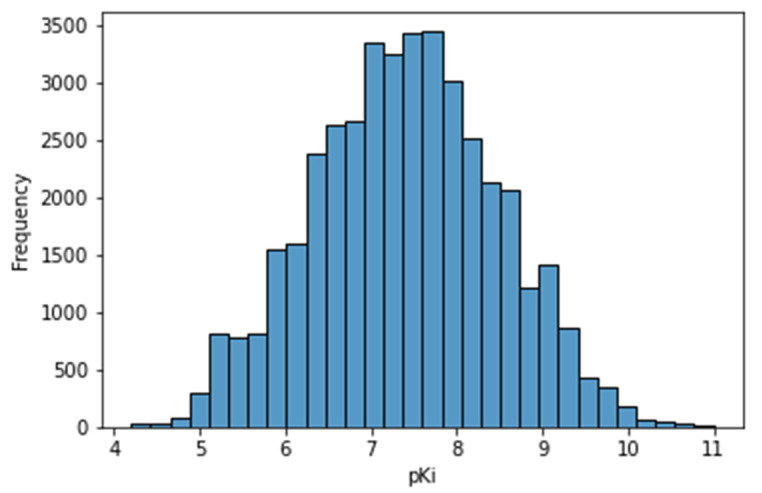
Histogram of pKi value in the curated database.

**Figure 4 pharmaceutics-13-01711-f004:**
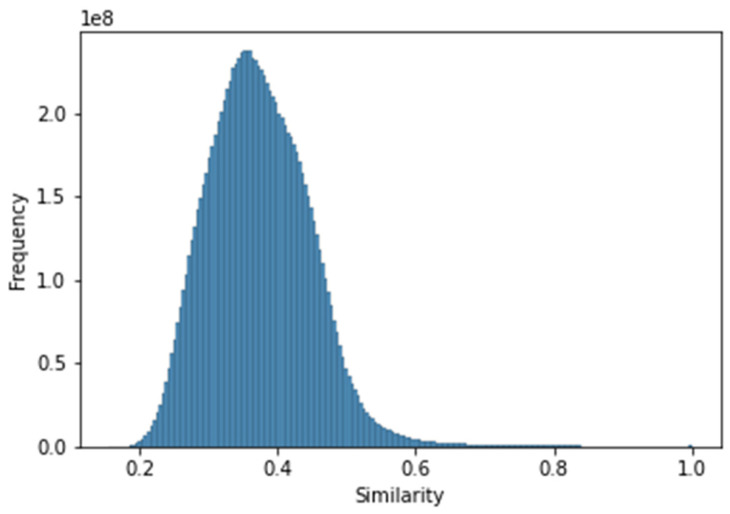
Histogram of similarity between compounds for the 5-HT1A curated database.

**Figure 5 pharmaceutics-13-01711-f005:**
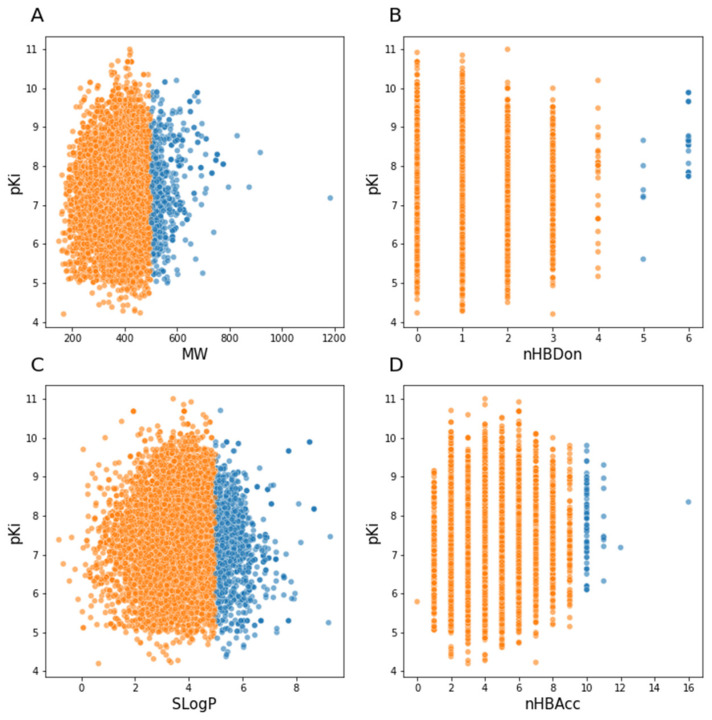
Scatterplots of properties from the rule of five and pKi value. (**A**) MW, molecular weight; (**B**) nHBDon, number of H-bond donors; (**C**) SLogP, *n*-octanol/water partition coefficient; (**D**) nHBAcc, number of H-bond acceptors.

**Figure 6 pharmaceutics-13-01711-f006:**
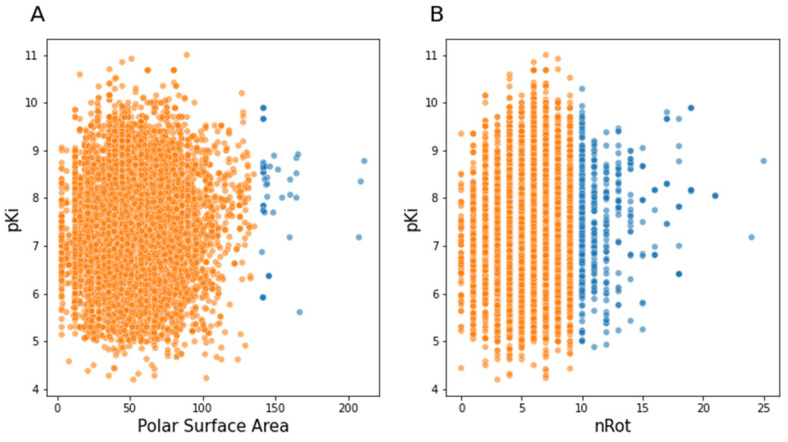
Scatterplots of pKi value and advanced properties of the rule of five: (**A**) polar surface area; (**B**) number of rotatable bonds (nRot).

**Figure 7 pharmaceutics-13-01711-f007:**
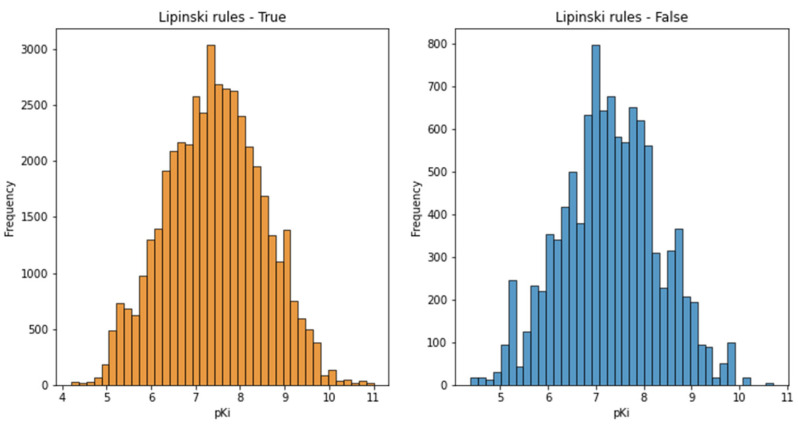
Histograms of pKi values for compounds that did and did not fulfill the rule of five.

**Figure 8 pharmaceutics-13-01711-f008:**
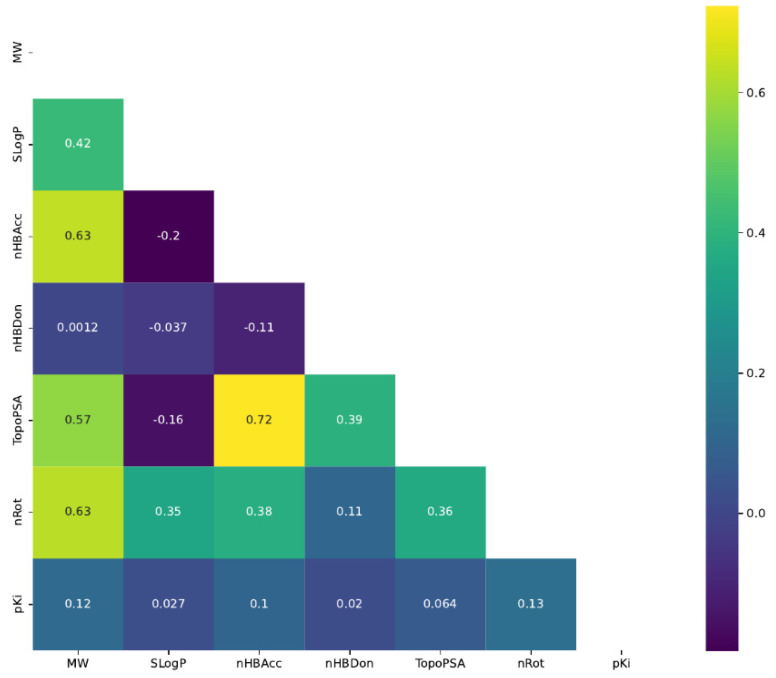
Correlation matrix of six chemical descriptors and pKi. Pearson correlation coefficients are displayed on the heatmap. MW, molecular weight; nHBDon, number of H-bond donors; SLogP, *n*-octanol/water partition coefficient; nHBAcc, number of H-bond acceptors; TopoPSA, polar surface area; nRot, number of rotatable bonds; pKi, affinity value for 5-HT1A serotonin receptor.

**Figure 9 pharmaceutics-13-01711-f009:**
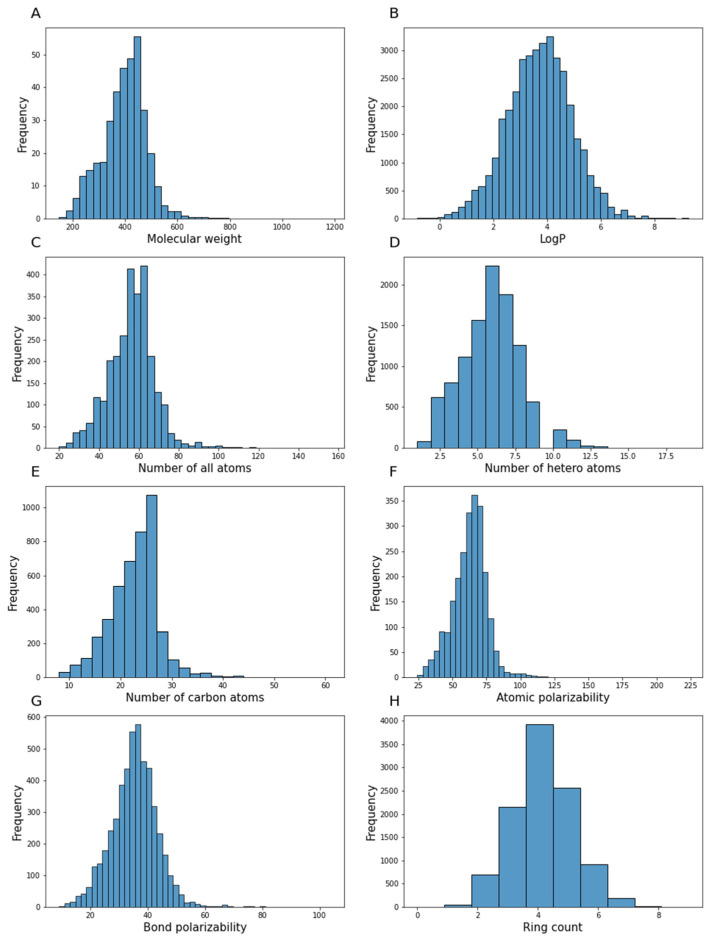
Example histograms of Mordred descriptor: (**A**) molecular weight, (**B**) LogP value, (**C**) number of all atoms, (**D**) number of heteroatoms, (**E**) number of carbon atoms, (**F**) atomic polarizability, (**G**) bond polarizability, and (**H**) ring count in the curated database of 5-HT1A receptors.

**Figure 10 pharmaceutics-13-01711-f010:**
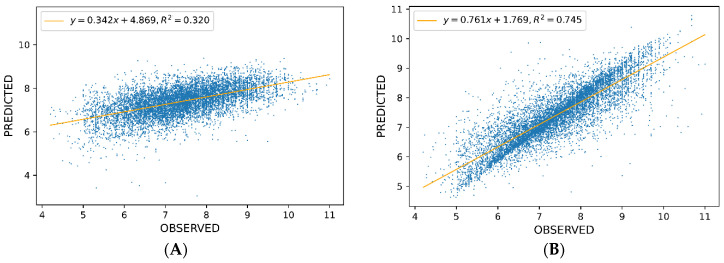
Predicted vs. observed values for (**A**) linear model and (**B**) AutoML model.

**Figure 11 pharmaceutics-13-01711-f011:**
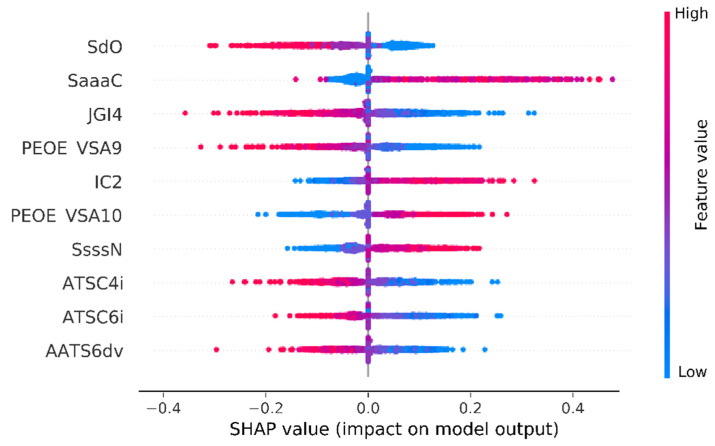
Aggregated feature plot ordered according to the mean absolute SHAP values. SdO—sum of dO; SaaaC—sum of aaaC; JGI4—four-ordered mean topological charge; PEOE VSA9—MOE charge VSA descriptor 9; IC2—two-ordered neighborhood information content; PEOE VSA10—MOE charge VSA descriptor 10; SsssN—sum of sssN; ATSC4i—centered Moreau–Broto autocorrelation of lag 4 weighted by ionization potential; ATSC6i—centered Moreau–Broto autocorrelation of lag 6 weighted by ionization potential; AATS6dv—averaged Moreau–Broto autocorrelation of lag 6 weighted by valence electrons [46].

**Figure 12 pharmaceutics-13-01711-f012:**
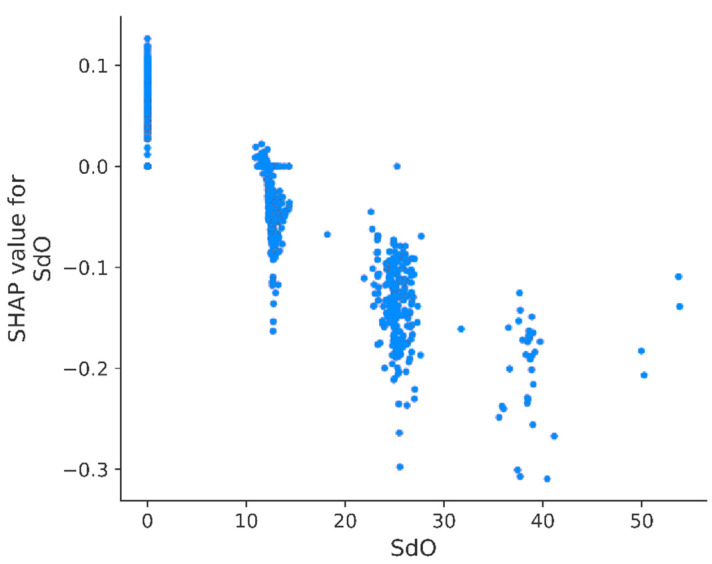
Calculated SHAP values for individual records as a function of SdO descriptor value.

**Figure 13 pharmaceutics-13-01711-f013:**
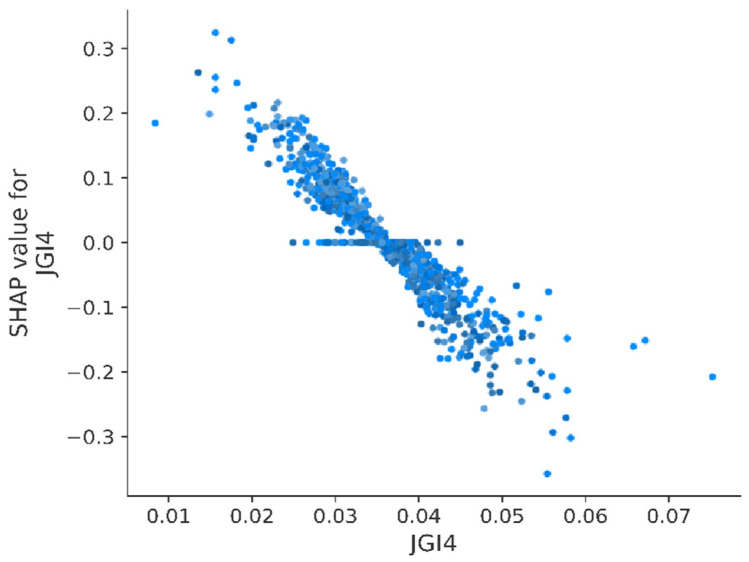
Calculated SHAP values for individual records as a function of JGI4 descriptor value.

**Table 1 pharmaceutics-13-01711-t001:** Summary of compounds from the curated database fulfilling the rule of five.

Feature	Number of Compounds Fulfilling the Rule	Percentage of Curated Database Fulfilling the Rule
Molecular weight	8691	92.07%
LogP	8190	86.76%
H-bond acceptors	9356	99.11%
H-bond donors	9403	99.61%
**Complete Lipinski’s rules**	**7735**	**81.94%**
Polar Surface Area	9382	99.39%
Rotatable Bonds	8848	93.73%
**Complete Veber et al.’s rules**	**7431**	**78.72%**

**Table 2 pharmaceutics-13-01711-t002:** Examples of approved drugs present in the curated database.

No.	Drug	pKi	The Rule of Five
1.	Amoxapine	6.66	✓
2.	Apomorphine	6.53	✓
3.	Aripiprazole	8.77	✓
4.	Bromocriptine	7.62	MW over limit
5.	Buspirone	7.46	✓
6.	Cariprazine	8.59	✓
7.	Clozapine	6.80	✓
8.	Haloperidol	5.77	✓
9.	Lisuride	9.40	✓
10.	Lofexidine	6.90	✓
11.	Naluzotan	8.29	✓
12.	Olanzapine	5.57	✓
13.	Pergolide	8.40	✓
14.	Pindolol	7.65	✓
15.	Quetiapine	6.52	✓
16.	Risperidone	6.72	✓
17.	Sumatriptan	6.64	✓
18.	Vilazodone	9.70	✓
19.	Vortioxetine	8.02	✓
20.	Ziprasidone	8.70	✓

**Table 3 pharmaceutics-13-01711-t003:** Top 20 variables from 216 selected during feature selection.

Chemical Descriptor	Description	Relative Importance	Original Input No
AATSC5d	Averaged and centered Moreau–Broto autocorrelation of lag 5 weighted by sigma electrons	1.000	364
ATSC4d	Centered Moreau–Broto autocorrelation of lag 4 weighted by sigma electrons	0.930	255
SpMAD_Dzi	Spectral mean absolute deviation from Barysz matrix weighted by ionization potential	0.912	761
SlogP_VSA1	MOE type descriptors using Wildman–Crippen LogP and surface area contribution	0.614	1080
nBondsS	Number of single bonds in non-Kekulized structure	0.556	773
BalabanJ	Balaban’s J index	0.517	665
SsssCH	Sum of sssCH	0.494	921
SssO	Sum of ssO	0.415	940
SssCH2	Sum of ssCH_2_	0.407	917
nBase	Basic group count	0.384	5
Xch-5dv	5-ordered chi chain weighted by valence electrons	0.369	800
SaaaC	Sum of aaaC	0.346	925
MATS5d	Moran coefficient of lag 5 weighted by sigma electrons	0.327	469
SLogP	Wildman–Crippen LogP	0.299	1229
PEOE_VSA9	MOE type descriptors using Gasteiger charge and surface area contribution	0.298	1067
AATS6dv	Averaged Moreau–Broto autocorrelation of lag 6 weighted by valence electrons	0.291	140
SdO	Sum of dO	0.281	939
SaasN	Sum of aasN	0.280	936
AATS4i	Averaged Moreau–Broto autocorrelation of lag 4 weighted by ionization potential	0.278	228

## Data Availability

(1) Final model file: 5-HT1A_model.zip, stacked ensemble model compatible with H_2_O v. 3.26.0.8. Available online: https://u:/r/personal/natalia_czub_doctoral_uj_edu_pl/Documents/PhD/github/5-HT1A_model.zip?csf=1&web=1&e=XGUDGA (accessed on 13 October 2021); (2) database file: 5-HT1A_mordred_curated_database.txt, curated database consisting of chemical descriptors with corresponding pKi values for 9440 compounds. Available online: https://ujchmura-my.sharepoint.com/:t:/r/personal/natalia_czub_doctoral_uj_edu_pl/Documents/PhD/github/5-HT1A_mordred_curated_database.txt?csf=1&web=1&e=bugP0I (accessed on 13 October 2021).

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
