# Peer review of "Curated Database and Preliminary AutoML QSAR Model for 5-HT1A Receptor"

_pharmaceutics, 2021, doi:10.3390/pharmaceutics13101711_

Round 1

Reviewer 1 Report

This is an interesting paper, related to the construction of a robust model with a good database model predicting serotonergic activity towards 5-HT1A receptor. The study was well conducted and the results are good. I consider that this article can be accepted in its current form.

Author Response

We would like to thank the Reviewer for the kind comments about the article. We do believe that provided results might speed up the process of search for new drugs.

Reviewer 2 Report

The present study is focused on the serotonin receptor 5-HT1A, which is a
therapeutic target in many CNS diseases (depression, schizophrenia, anxiety, and cognitive disorders), and might be utilized in and the treatment of serotonin-dependent neoplastic diseases in the future. 

The developed model can efficiently predict the affinity value for new molecules towards the 5-HT1A receptor based on their structure. Usage of
this model in screening processes can significantly improve the process of searching for new drugs in the field of mental diseases and anticancer therapy.

The serotonin problem is very complex. It is involved in a very large number of functions in the human body, with a broad relationship with other neurotransmitters. Therefore, all studies focused on 5-HT should be welcomed to try to systematize their subsequent studies. 

Based on these reasoning, this article could be accepted as a means to design active structures within this difficult research field.

Author Response

(The authors gave the same response as above.)

Reviewer 3 Report

The authors, paper entitled: Curated database and preliminary AutoML QSAR model for 5-HT1A receptor, sounds very promising since the team of scientist has made an interesting approach to execute classical drug searching methods, such as QSAR. Also, the background is provided in a good way, allowing the readers to perfectly understand it. Moreover, the methods are covered in detail, making it more reproductible. Finally, the results, discussions and conclusions are very satisfactory and they fitted with the purpose of this study. Then, I believe that AutoML QSAR could be a promising alternative to search for novel drugs or bioactive molecules. Regarding the whole manuscript, I strongly recommend it for publication in its current form.

Author Response

We would like to thank the Reviewer for the kind comments about the article. We do believe that provided results might speed up the process of search for new drugs

Reviewer 4 Report

The manuscript is considered a new addition in the field

The manuscript contains some errors in English syntax and spelling that require corrections 

Where the graphical abstract 

Author Response

We would like to thank the Reviewer for the kind comment about the article. We have made numerous improvements in the text, and we prepared graphical abstract